# Garcinol Promotes the Formation of Slow-Twitch Muscle Fibers by Inhibiting p300-Dependent Acetylation of PGC-1α

**DOI:** 10.3390/ijms24032702

**Published:** 2023-01-31

**Authors:** Weilei Yao, Baoyin Guo, Taimin Jin, Zhengxi Bao, Tongxin Wang, Shu Wen, Feiruo Huang

**Affiliations:** 1Department of Animal Nutrition and Feed Science, College of Animal Science and Technology, Huazhong Agricultural University, Wuhan 430070, China; 2The Cooperative Innovation Center for Sustainable Pig Production, Wuhan 430070, China

**Keywords:** skeletal muscle, fiber type, garcinol, p300, PGC-1α

## Abstract

The conversion of skeletal muscle fiber from fast-twitch to slow-twitch is crucial for sustained contractile and stretchable events, energy homeostasis, and anti-fatigue ability. The purpose of our study was to explore the mechanism and effects of garcinol on the regulation of skeletal muscle fiber type transformation. Forty 21-day-old male C57/BL6J mice (n = 10/diet) were fed a control diet or a control diet plus garcinol at 100 mg/kg (Low Gar), 300 mg/kg (Mid Gar), or 500 mg/kg (High Gar) for 12 weeks. The tibialis anterior (TA) and soleus muscles were collected for protein and immunoprecipitation analyses. Dietary garcinol significantly downregulated (*p* < 0.05) fast myosin heavy chain (MyHC) expression and upregulated (*p* < 0.05) slow MyHC expression in the TA and soleus muscles. Garcinol significantly increased (*p* < 0.05) the activity of peroxisome proliferator-activated receptor gamma co-activator 1α (PGC-1α) and markedly decreased (*p* < 0.05) the acetylation of PGC-1α. In vitro and in vivo experiments showed that garcinol decreased (*p* < 0.05) lactate dehydrogenase activity and increased (*p* < 0.05) the activities of malate dehydrogenase and succinic dehydrogenase. In addition, the results of C2C12 myotubes showed that garcinol treatment increased (*p* < 0.05) the transformation of glycolytic muscle fiber to oxidative muscle fiber by 45.9%. Garcinol treatment and p300 interference reduced (*p* < 0.05) the expression of fast MyHC but increased (*p* < 0.05) the expression of slow MyHC in vitro. Moreover, the acetylation of PGC-1α was significantly decreased (*p* < 0.05). Garcinol promotes the transformation of skeletal muscle fibers from the fast-glycolytic type to the slow-oxidative type through the p300/PGC-1α signaling pathway in C2C12 myotubes.

## 1. Introduction

Skeletal muscle is composed of muscle fibers with different physiological, biochemical and metabolic properties [1,2]. Adult skeletal muscle fibers are generally classified into two major groups: slow-twitch (type I) and fast-twitch (type II) muscle fibers. Type I muscle fibers express myosin heavy chain I (MyHC I), and type II muscle fibers express MyHC IIa, MyHC IIx and MyHC IIb [3]. Muscle function is related to the composition of muscle fiber type and can be improved by modulation of this composition [4]. Slow-twitch muscle fibers are rich in mitochondria and have high endurance and strong resistance to fatigue. Fast-twitch muscle fibers mainly utilize glycolytic metabolism and have low endurance, fewer mitochondria, and low fatigue resistance. Muscle fibers are not static and may be transformed under external conditions during skeletal muscle development. Skeletal muscle fiber proportion can be influenced by nutritional status, energy metabolism level and exercise. It was found that malnutrition in piglets led to a significant increase in the proportion of type IIb muscle fibers in the longest dorsal muscle and a significant decrease in the proportion of type IIa muscle fibers [5]. In addition, endurance exercise increased PGC-1 expression in skeletal muscle to promote type I muscle fiber content [6]. Therefore, regulation of the transformation of skeletal muscle fibers from the fast-twitch type to slow-twitch type through nutritional strategies has attracted the attention of researchers.

It is well known that an increase in the proportion of slow muscle fibers is beneficial to improve muscle endurance. Recently, there have been many studies on slow muscle fiber promotion in mice through dietary supplementation with nutrients, such as resveratrol and apple polyphenols [7,8,9]. These nutrients contain a variety of polyphenols. Polyphenols can increase Sirt1 activity in skeletal muscle, which in turn promotes the activation of PGC-1α [10]. It has been speculated that natural plant extracts may regulate the conversion of muscle fiber types by regulating PGC-1α activity. Moreover, PGC-1α can upregulate the expression of oxidative muscle fiber-specific genes, which can promote the transformation from glycolytic muscle fiber to oxidative muscle fiber [11,12]. Studies have shown that PGC-1α can be regulated not only by Sirt1 but also by CREB-binding protein (CBP)/p300 [13,14]. Wallberg et al. found that PGC-1α was modified by the acetylation of acetyltransferase p300, which led to the loss of transcriptional activity of PGC-1α [15]. Therefore, decreasing the acetylation of PGC-1α may regulate the type composition of muscle fibers.

Garcinol is the major component of the *Garcinia indica* (*G. indica*) fruit rind, extensively used as a traditional treatment for gastric ailments and skin irritation. Garcinol is regarded as an extremely potent natural inhibitor of p300, which has been observed to activate the glycolysis pathway through the acetylation of the glycolytic enzymes [16,17]. Furthermore, numerous studies have shown that garcinol has a wide range of biological activities, including those useful for treating obesity and diabetes and anti-inflammatory and antioxidant effects, preventing early atherosclerotic events, and extending the life span of various species [18,19,20]. However, the effect of garcinol on p300-regulated muscle fiber-type conversion is unknown. The purpose of our study was to investigate the effects of dietary supplementation with garcinol on the muscle fiber type composition and growth performance in mice to provide theoretical and experimental bases for understanding the biological role of garcinol.

## 2. Results

### 2.1. Effect of Garcinol Supplementation on Endurance Exercise Capacity and Muscle Fatigue in Mice

As shown in Figure 1, feed intake was not affected by garcinol treatment. The body weight, average daily weight gain (ADG), average daily feed intake (ADFI) and the ratio of weight gain to feed intake (F/G) measures of the garcinol-treated mice were similar to those of the control (Figure 1A–E). The Mid and High Gar groups increased (*p* < 0.05) muscle grip strength (Figure 1F), swimming time (Figure 1G), and low-speed running time (Figure 1H). However, the high-speed running time was unchanged by garcinol supplementation (Figure 1I). In addition, there was no change in the relative weights of the TA and soleus muscles from C57/BL6J mice fed a diet supplemented with 500 mg/kg garcinol (Figure 1J). Moreover, the weight of different tissues from the mice administered with garcinol was unchanged. In summary, garcinol supplementation had no effect on the growth of mice.

### 2.2. Effect of Garcinol Supplementation on the Muscle Fiber Type-Related Protein Expression in Mice

MyHC IIb fast fibers of the TA were decreased by treatment with 500 mg/kg garcinol, while MyHC I slow fibers of the soleus were significantly increased (*p* < 0.05, Figure 2A,E). Garcinol significantly decreased the mRNA expression of the fast-twitch fiber marker Troponin I-F and increased the mRNA expression of the slow-twitch fiber markers myoglobin and Troponin I-S (*p* < 0.05, Figure 2B,F). Compared with that in the Low Gar and control groups, the protein expression of slow MyHC in the High and Mid Gar groups was higher (*p* < 0.05). The Low Gar group had expression levels of fast MyHC and p300 similar to those of the control group (*p* ≥ 0.05), while the Mid and High groups had lower (*p* < 0.05) expression levels than the control and Low Gar groups. Moreover, the expression of PGC-1α and Sirt1 in the TA and soleus muscles was not significantly different among all groups (*p* ≥ 0.05) (Figure 2C–H).

### 2.3. Effect of Garcinol Supplementation on the Acetylation Levels of PGC-1α in Muscle

As shown in Figure 3A, the enzyme activity of p300 tended to decrease (*p* = 0.06) in the Low Gar group, while its activity was significantly decreased (*p* < 0.05) in the Mid and High Gar groups compared with that in the control group (Figure 3A,D). Compared with that in the Low Gar and control groups, the PGC-1α activity in the High and Mid Gar groups was obviously increased (*p* < 0.05) (Figure 3B,E). Additionally, the acetylation level of PGC-1α in the garcinol-fed mice was reduced significantly (*p* < 0.05) compared with that in the control group mice (Figure 3C,F).

### 2.4. Effect of Garcinol Supplementation on Antioxidant Capacity and Metabolic Enzyme Activities

As shown in Table 1, dietary garcinol supplementation significantly increased (*p* < 0.05) the GSH content and the activities of SOD, GPx and CAT in mice serum, TA and soleus muscle, and decreased the MDA content in mice TA and soleus muscle. The activity levels of SDH and MDH in the TA and soleus muscles of the Mid and High Gar groups were significantly increased (*p* < 0.05) by garcinol supplementation, whereas LDH activity was decreased (*p* < 0.05) in the Low Gar and control groups (Figure 4A,B). The activity levels of SDH and MDH in C2C12 myotubes were also significantly increased (*p* < 0.05) by garcinol treatments, whereas the LDH activity was decreased (*p* < 0.05) after treatment with garcinol (Figure 4C). In addition, Garcinol significantly increased (*p* < 0.05) mitochondrial DNA content and the activity of the complex in C2C12 myotubes (Figure 4D,E). Garcinol significantly increased (*p* < 0.05) oxygen consumption in C2C12 myotubes, but reduced glycolytic metabolism (Figure 4F,G).

### 2.5. Effect of Garcinol Supplementation on Myofiber Composition

To investigate the role of garcinol in skeletal muscle fiber type-related protein expression, we performed a Western blotting analysis in C2C12 myotubes. As shown in Figure 5A, compared with the control, the protein expression of slow MyHC was up-regulated (*p* < 0.05) by garcinol in C2C12 myotubes, whereas the fast MyHC protein was down-regulated by garcinol treatments.

### 2.6. Garcinol Inhibits the p300-Dependent Acetylation of PGC-1α in C2C12 Myotubes

C2C12 myotubes were transfected with the indicated K (lysine) acetyltransferase (KAT) constructs in combination with expression vectors containing p300. As expected, p300 expression was increased (*p* < 0.05) by 60%, while transfection with siRNAs targeting p300 and PGC-1α in C2C12 myotubes decreased the protein expression by ∼70% (*p* < 0.05, Figure 6A). The results also showed that overexpression of p300 decreased (*p* < 0.05) the protein expression level of slow MyHC, the mRNA expression levels of myoglobin and Troponin I-S, and the activity level of PGC-1α. The protein expression level of fast MyHC, the mRNA expression of Troponin I-F mRNA, and the acetylation level of PGC-1α were significantly higher (*p* < 0.05) in the case of overexpression of p300.However, treatment of C2C12 myotubes overexpressing p300 with garcinol reversed these effects (*p* < 0.05, Figure 6B–E). Furthermore, knocking down PGC-1α by siRNA significantly reduced (*p* < 0.05) the protein expression levels of slow MyHC and the mRNA expression of myoglobin and troponin I-S, and increased (*p* < 0.05) the mRNA expression of troponin I-F (Figure 6F,G). There was no significant change in PGC-1α knockdown in the C2C12 myotubes treated with garcinol. Knocking down p300 increased (*p* < 0.05) the protein expression levels of slow MyHC, the mRNA expression of myoglobin and troponin I-S, the activity level of PGC-1α, and decreased (*p* < 0.05) the protein expression levels of fast MyHC and the mRNA expression of troponin I-F. Moreover, the acetylation level of PGC-1α in C2C12 myotubes with p300 knocked down was decreased (*p* < 0.05, Figure 6H,I).

## 3. Discussion

In this investigation, the potential of garcinol to stimulate slow oxidative fiber formation was evaluated. We found that the expression of slow MyHC and PGC-1α activity can be upregulated by garcinol treatment in vitro and *in vivo*. Mechanistically, we proved that the expression of slow MyHC was upregulated by garcinol through the p300/PGC-1α pathway.

In the 1980s, garcinol was first extracted from *Garcinia indica* (*G. indica*, also known as kokum) of the Western Ghats in India [21]. Many studies have indicated the anti-obesity, anti-inflammatory, antioxidant, and antiglycation effects of garcinol [22,23]. Similar to findings in previous studies, the body weight and food intake were not affected by treatment with 500 ppm garcinol for 12 weeks in the current study. Generally, slow muscles have higher endurance and greater resistance to fatigue than fast muscles. We found that garcinol dose-dependently increased muscle grip strength, swimming time, and low-speed running time. However, the high-speed running time was unchanged by garcinol supplementation, which indicates garcinol may specifically improve endurance exercise performance, but not explosive exercise performance. Our data showed that dietary garcinol supplementation improved endurance exercise performance and attenuated skeletal muscle fatigability, accompanied by upregulated slow MyHC expression. There is evidence that antioxidant enzymes are more in slow fibers than fast fibers [24]. Our results also showed that garcinol significantly increased antioxidant enzyme activity in blood and muscle in mice. There is evidence that some antioxidants can directly affect mitochondrial activity by modulating PGC-1α/PPAR-α signaling pathway [25]. Moreover, the quantity of fast MyHC-positive cells was decreased, while the quantity of slow MyHC-positive cells was increased, as indicated by the immunofluorescence assay results. Therefore, the results show that the transformation of muscle fibers from the fast-twitch type to the slow-twitch type was accelerated by garcinol. The type of skeletal muscle fiber can be classified on the basis of the characteristics of energy metabolism, and the transformation of skeletal muscle fiber types can be indirectly affected by enzyme activity in energy metabolism [26]. For example, glycolytic enzyme (LDH) has a higher activity in glycolytic fibers, while oxidases (MDH and SDH) have higher activity in oxidative fibers. Our results indicate that the activity of LDH was decreased and the activities of MDH and SDH were increased in mice, which supports the supposition that the transformation of muscle fiber types is influenced by garcinol. 

According to previous research, the transformation of muscle fiber types and the energy metabolism of muscle tissues are related to the PGC-1α signaling pathway [27,28]. Several previous studies on transgenic mice concluded that the conversion of skeletal muscle fibers from the fast MyHC type to the slow MyHC type is caused by specific overexpression of PGC-1α [29]. Additional studies found that PGC-1α transgenic animals have a higher proportion of slow MyHC muscle fibers, and the transformation of muscle fiber types to slow MyHC was promoted [30]. In addition, the mRNA expression of glycolytic fast muscle-related intermediates MyHC-IIx and MHC-IIb was decreased and the mRNA expression of oxidative slow muscle-associated MyHC-Ⅰ was increased by the overexpression of PGC-1α [31]. Our results showed that the expression and activity of p300 in the TA and soleus muscles were decreased by garcinol supplementation. Moreover, the muscle fiber type results indicated that the transcription level of PGC-1α was upregulated by garcinol in vivo and in vitro, which may have modulated slow MyHC-type fiber transcriptional activity. Thus, we hypothesized that garcinol can regulate the PGC-1α signaling pathway to regulate skeletal muscle fiber transformation.

To confirm whether garcinol-induced changes in molecules were related to increased slow MyHC muscle fibers, we assessed PGC-1α activity in C2C12 myotubes. Garcinol can inhibit acetylase, which seemingly regulates myofiber transformation. According to previous research, garcinol could induce apoptosis of cell lines of pancreatic cancer, prostate cancer and leukemia, and it can quickly inactivate p300 by inhibiting the NFκB-DNA interaction [32,33]. In the present study, garcinol increased PGC-1α activity and increased slow-twitch muscle fibers in mice and C2C12 myotubes, which was demonstrated by the reduction in PGC-1α acetylation induced by garcinol treatment, so we propose that the p300-PGC-1α pathway may participate in the improvement of myofiber transformation in mice fed a diet supplemented with garcinol. Specifically, p300 could increase PGC-1α acetylation when p300 was overexpressed, which was observed to be inactivated after garcinol intervention in vitro. Overexpression of siRNAs inhibits p300 activity in cultured C2C12 myotubes, indicating that nongarcinol-mediated inhibition of p300 has similar effects. In addition, the expression of fast muscle fibers and biomarkers increased under PGC-1α knockdown conditions. However, the treatment of garcinol has no effect on weakening the expression of fast-twitch muscle fibers in C2C12 myotubes with knockdown PGC-1α. Therefore, we found that the change from fast-twitch to slow-twitch skeletal muscle fibers was related to higher PGC-1α activity, which fit the garcinol working model. Our results revealed that garcinol may increase slow-twitch muscle fibers via inhibiting p300-induced PGC-1α acetylation in mice and C2C12 myotubes. All of these results show that p300 is related to an increase in slow muscle fibers, but the precise regulatory mechanism needs to be studied further. 

## 4. Materials and Methods

### 4.1. Animals, Treatments, and Sample Collection

A total of forty 3-week-old male C57/BL6J mice with an average body weight of 12.75 ± 0.21 g were divided into 4 treatments (10 mice per treatment, License No. SCXK Gui 2014-0002). The environmental conditions of the mouse breeding facility included a 12 h:12 h light:dark cycle, relative humidity of 50% and temperature of 25 °C. During the experiment, the mice were given free access to water and their diet. The Jiangsu Xietong Biotechnology Corporation in China provided the control diet used in this research. The nutrient recipe for the mouse fodder was composed in accordance with the National Criterion of China, GB14924.3-2010 (Appendix A). Mice were randomly allocated into four groups (n = 10) in which mice were fed a control diet containing 0 mg/kg garcinol (control group), 100 mg/kg garcinol (Low Gar group), 300 mg/kg garcinol (Mid Gar group), and 500 mg/kg garcinol (High Gar group). Garcinol obtained from Biomol/Enzo Life Sciences International, Inc., was added to the control diet powder. The experimental period was 12 weeks. Mice were singularly housed in order to measure the food intake in home cages. Food intake and body weight were monitored every day until the end of the study. The average daily feed intake (ADFI), average daily weight gain (ADG) and the ratio of weight gain to feed intake (G/F) were calculated. At the end of the experiment, blood samples were collected after a 12 h fast. All the mice were killed under CO_2_ anesthesia. The tibialis anterior (TA) and soleus muscles were collected for analysis.

### 4.2. Strength and Exercise Endurance

Thirty minutes after the last treatment, mice were made to perform endurance and strength exercises. Mouse had maximum muscle force measured three times by a grip strength meter (BIO-GS3, Bioseb/France), and the mean maximum strength was used for data analysis. The mice underwent a weight-loaded swimming test using lead wire, which was 5% of each mouse’s body weight and attached to the tail root of the mice. The mice were tested individually under the same conditions (30 ± 1 °C, 50 cm depth). When the mice failed to keep their head above the water continuously over a 10 s time frame and showed a lack of coordinated movements, the time to exhaustion was immediately recorded [34,35]. The mice performed a treadmill-running test on the FT-200 Animal treadmill at an initial velocity of 10 m/min for 10 min in order to keep mice sober. Then, velocity was increased by 5 m/min every 2 min until 40 m/min in high-speed running tests, and 1 m/min every 3 min in low-speed running tests. The above tests refer to the previous study [36].

### 4.3. Cell Culture and Treatment

C2C12 myoblasts were cultured in Dulbecco’s modified Eagle’s medium (DMEM) (Invitrogen, Carlsbad, CA, USA) supplemented with 10% fetal bovine serum (FBS) (Gibco, Paisley, Scotland, UK), 100 mg/L streptomycin and 100 U/mL penicillin (Gibco) at 37 °C in a 5% CO_2_ atmosphere. When the C2C12 myoblasts reached 80% confluence, cells were moved to a differentiation medium (DM) containing DMEM and 2% horse serum (Gibco). The C2C12 cells were seeded in 12-well plates. The cells were treated with 20 μM garcinol after 6 days of differentiation. The cells were then treated for 3 days prior to harvesting (Appendix A). For the following mechanistic studies, after 6 days of differentiation, the cells were treated with 20 μM garcinol (Appendix A), p300 siRNA (si-p300) (50 nM) or PGC-1α (si-PGC-1α) (GenePharma, Shanghai, China). Plasmids for pcDNA3-FLAG-tagged p300 were gifts from Wang et al. The transfection reagent was incubated with plasmid DNA constructs in Opti-MEM (Invitrogen) at room temperature for 20 min. Then, this transfection mixture was added to the proliferating cell culture and incubated for 5 h at 37 °C. Twenty-four hours after transfection, the medium was replenished with fresh differentiation medium. C2C12 myotubes were harvested for further analysis.

### 4.4. Western Blotting and Immunoprecipitation

Western blot analyses of the pertinent samples were performed as previously described [37]. The primary antibodies, including anti-slow-MyHC (Cat.No.M8421), anti-fast-MyHC (Cat.No.M4276), anti-p300 (Cat.No. MA1-16608), anti-PGC-1α (Cat. No. 2178S), anti-Sirt1 (Cat.No. 8469S) and anti-β-actin (ACTB, Cat. No. sc-8432), used for the analyses are presented in Appendix A. Immunoprecipitation was carried out either by incubating FLAG/Myc beads with cell lysate at 4 °C for 3–4 h or by incubating the appropriate antibody with cell lysate for 2–3 h, followed by incubation with protein-A beads (Upstate). Standard Western blot procedures were used to perform a protein and tag analysis. For determining acetylation of proteins by Western blotting, 50 mM Tris (pH 7.5) with 10% (vol/vol) Tween-20 and 1% peptone (AMRESCO) were used for blocking, and 50 mM Tris (pH 7.5) with 0.1% peptone was used to prepare the primary and secondary antibodies [38].

### 4.5. Mitochondrial DNA Content and Respiratory Chain Complex Activity

Total cellular DNA was extracted from C2C12 cells with DNAzol reagent (Invitrogen, CA, USA) according to the manufacturer’s instructions. Mitochondrial DNA copy number was determined by quantification of four mitochondrial marker genes, including mitochondrially encoded ATP synthase membrane subunit 6 (ATPase6), cytochrome c oxidase subunit 2 (COX2), Mit-1000, and mitochondrial-encoded cytochrome b (mt-Cytb). The expression level of ATPase6, COX2, Mit-1000, and mt-Cytb was tested in quantitative real-time-PCR and normalized to an intron of the nuclear-encoded b-globin gene. The activity of mitochondrial complexes I (NADH-CoQ reducing enzyme), II (succinate coenzyme Q reducing enzyme), III (CoQ-cytochrome C reducing enzyme) and IV (cytochrome C oxidase) was determined by using the kits from Shanghai JamiGen Pharmaceutical Technology Co.

### 4.6. Enzymatic Activity Assay

C2C12 myotubes were taken using a scraper, lysed in PBS, centrifuged at 2000 rpm for 20 min and collected the liquid supernatant for further analysis. Cell protein concentrations were determined by BCA protein assay kit (Pierce, USA) and Nano-Drop ND 2000c Spectrophotometer (Thermo Scientific, USA). The lactate dehydrogenase (LDH, Catalog No. A020-2-1) activity, malate dehydrogenase (MDH, Catalog No. A021-2-1) activity, and succinic dehydrogenase (SDH, Catalog No. A022-1-1) activity in the TA muscle and soleus muscle of mice and in C2C12 myotubes were measured using commercial kits according to the manufacturers’ instructions (Nanjing Jiancheng Bioengineering Institute, Nanjing, China). Complex III ELISA kit and Complex IV ELISA kit (Shanghai Enzyme-linked Biotechnology Co., Ltd., Shanghai, China) were used to test Complex III and Complex IV concentrations in muscle tissue. HAT p300 ELISA kit (Catalog No. 50092, Anaspec, Fremont, CA, USA) was used to test p300 activity. The activities of the specific enzyme were defined as U/mg of protein units.

### 4.7. Seahorse Oxidative and Glycolytic Metabolic Assay

Cells were seeded into Seahorse XFe96 culture plates and differentiated for 6 days once reaching confluence. Following 72-h treatment with differentiation media or uOC at either 1 ng/mL, 10 ng/mL, or 100 ng/mL, the media was replaced with XF Assay Media obtained from Agilent Technologies (Santa Clara, CA). Baseline measurements of oxygen consumption rate (OCR) and extracellular acidification rate (ECAR) were recorded as indicators of basal oxidative metabolism and glycolytic metabolism, respectively. Following basal measurements, each well was infused with oligomycin (an inhibitor of ATP synthase) at a final concentration of 2 μM to induce maximal glycolytic metabolism. Cells were then exposed to carbonyl cyanide *p*-[trifluoromethoxy]-phenyl-hydrazone (FCCP) at 2 μM to uncouple electron transport and induce peak OCR. Maximal respiration measurements were followed by the injection of rotenone at 0.5 μM to reveal non-mitochondrial respiration. The Seahorse XFe96 Analyzer was run using a 6-min cyclic protocol command (mix for 3 min and measure for 3 min). Seahorse experiments were completed with 22–23 replicates per group.

### 4.8. PGC-1α Transcriptional Activity

The wild-type and mutant 3’UTR of PGC1-α were inserted into the psiCHECK™-2 vector (Promega, Madison, WI, USA) between the XhoI and NotI restriction sites at the 3’ end of the Renilla luciferase gene. Plasmids were sequenced (TsingKe Biotech, Beijing, China) to verify the correct insertion. For luciferase reporter analysis, C2C12 myotubes were co-transfected with the psiCHECK™-2 containing wild-type or mutant 3’ UTR of PGC1-α using Lipofectamine 3000 (Invitrogen, Waltham, MA, USA). Cells were harvested after 48 h of transfection and luciferase activity was measured using the Luciferase Assay System (Promega, Madison, WI, USA) according to the manufacturer’s instructions. Firefly luciferase was used as a normalization control.

### 4.9. Antioxidant Enzyme Activity Analysis

Total SOD (T-SOD, Catalog No. A001-1-1) activity, glutathione peroxidase (GPx, Catalog No. H545-1-1) activity, catalase (CAT, Catalog No. A007-1-1) activity, and malondialdehyde (MDA, Catalog No. A003-1-2) content were all determined by commercial assay kits which were purchased from Nanjing Jiancheng Bioengineering Institute (China).

### 4.10. RNA Isolation and Real-Time Quantitative PCR

Muscle tissue or C2C12 myotubes lysates were homogenized, and total RNA was extracted using TRIzol reagent (TaKaRa, Kusatsu, Japan) by standard techniques. Reverse transcription of mRNA was performed with a PrimeScript RT reagent Kit with gDNA Eraser (TaKaRa, China). Real-time PCR amplification reactions were carried out by Bio-Rad iQ5 with SYBR Premix Ex Taq TM II (TaKaRa) chemical detection under amplification conditions. The primer sequences used are listed in Appendix A. Relative expression levels of mRNAs were calculated using the 2^−ΔΔCt^ method, with 18S ribosomal RNA used as a control for normalization.

### 4.11. Statistical Analysis

Experimental data were analyzed by SAS 8.2 software (SAS Inst. Inc., Cary, NC, USA) and expressed as the mean ± SE (standard error). Two-group were compared using *t*-tests and multiple-group comparisons were performed using ANOVA and post hoc Tukey’s test. *p* < 0.05 was deemed to have statistical significance in all analyses.

## 5. Conclusions

Overall, our research revealed that garcinol can promote the conversion of muscle fiber from the glycolytic type to the oxidative type through the p300/PGC-1α pathway. Our findings enhance the comprehension of the effects of garcinol on muscle fiber type transformation and strongly indicate that garcinol can regulate the proportion of slow MyHC muscle fibers. The new treatment strategy provided by this study may potentially be useful in the treatment of muscle diseases and improve the reduction in endurance caused by oxidative muscle fiber deficiency. Additional studies are warranted to determine the safe dosage of garcinol in the human body, which may be helpful to maintain muscle energy homeostasis.

## Figures and Tables

**Figure 1 ijms-24-02702-f001:**
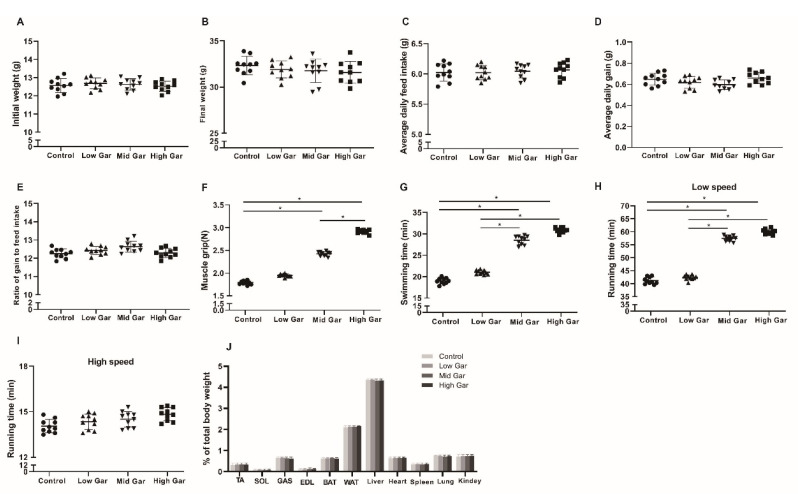
Effects of dietary garcinol on the Initial weight (**A**), Final weight (**B**), Average daily feed intake (**C**), Average daily gain (**D**), Ratio of gain to feed intake (**E**), Muscle grip (**F**), Swimming time (**G**), low-speed running time (**H**), high-speed running time (**I**) and the tissues weight (**J**) in mice fed 0–500 mg/kg garcinol for 12 weeks. Data are shown as the mean ± SE, n = 10. * Different from the control group, *p* < 0.05.

**Figure 2 ijms-24-02702-f002:**
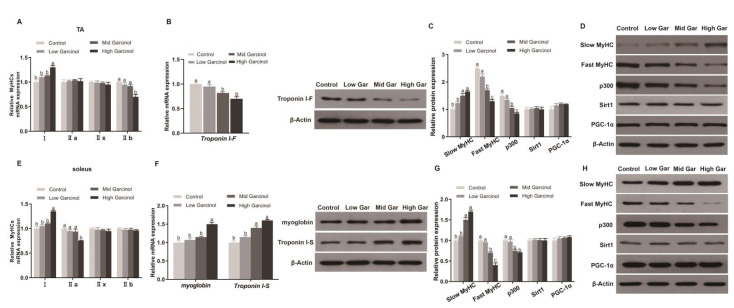
Real-time PCR was performed to quantify the relative mRNA expression of MyHC in TA (**A**) and soleus (**E**) muscle. Effects of dietary garcinol on the relative mRNA and protein expression of Troponin I-F in TA muscle (**B**), myoglobin and Troponin I-S in soleus muscle (**F**). Slow MyHC, Fast MyHC, p300, Sirt1 PGC-1α protein expression in TA (**C**,**D**) and soleus (**G**,**H**) muscle. Data are shown as the mean ± SE, n = 10. Means for a given protein without a common letter differ, *p* < 0.05.

**Figure 3 ijms-24-02702-f003:**
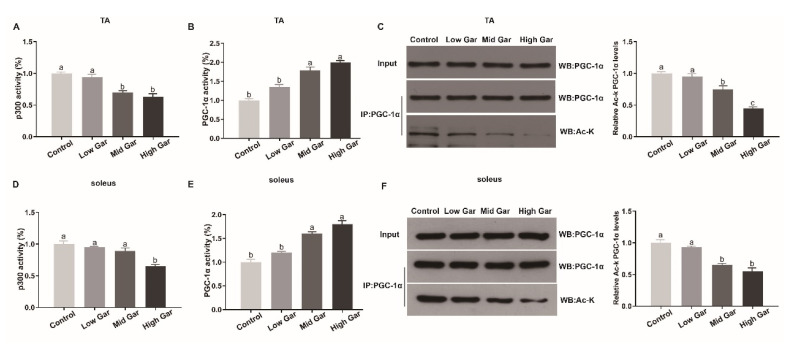
Effects of dietary garcinol on the activity of p300 (**A**), PGC-1α (**B**), acetylation level of PGC-1α (**C**) in TA muscle and the activity of p300 (**D**), PGC-1α (**E**), acetylation level of PGC-1α (**F**) in soleus muscle fed 0–500 mg/kg garcinol for 12 weeks. Data are shown as the mean ± SE, n = 10. Means for a given protein without a common letter differ, *p* < 0.05.

**Figure 4 ijms-24-02702-f004:**
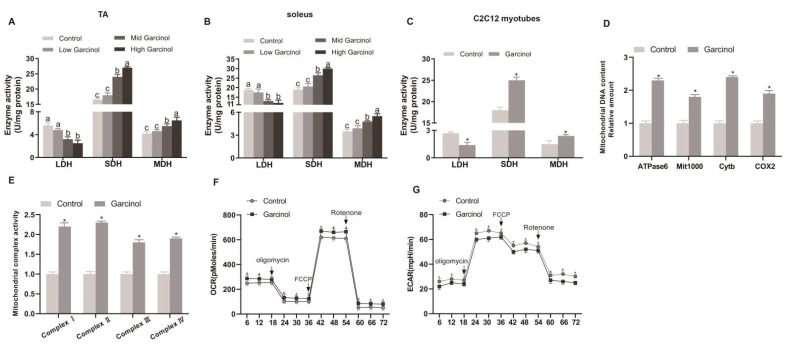
Effects of dietary garcinol on the activity of lactate dehydrogenase (LDH), succinic dehydrogenase (SDH), and malate dehydrogenase (MDH) in TA (**A**), soleus (**B**) muscle of mice and C2C12 myotubes (**C**), respectively. (**D**,**E**) Quantification of mitochondrial DNA contents and complex activity. (**F**,**G**) Effect of garcinol on oxygen consumption rate and extracellular acidification rate. Data are shown as the mean ± SE, n = 10. * Different from the control group, *p* < 0.05. Means without a common letter differ, *p* < 0.05.

**Figure 5 ijms-24-02702-f005:**
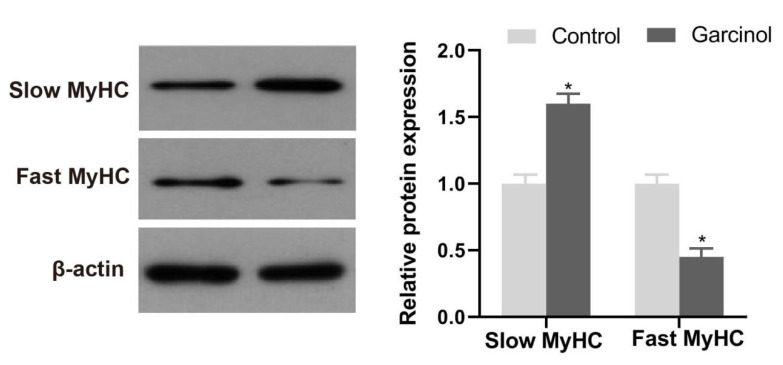
Effect of garciniol on slow MyHC and fast MyHC expression in C2C12 myotubes. Data are shown as the mean ± SE, n = 10. * Different from the control group, *p* < 0.05.

**Figure 6 ijms-24-02702-f006:**
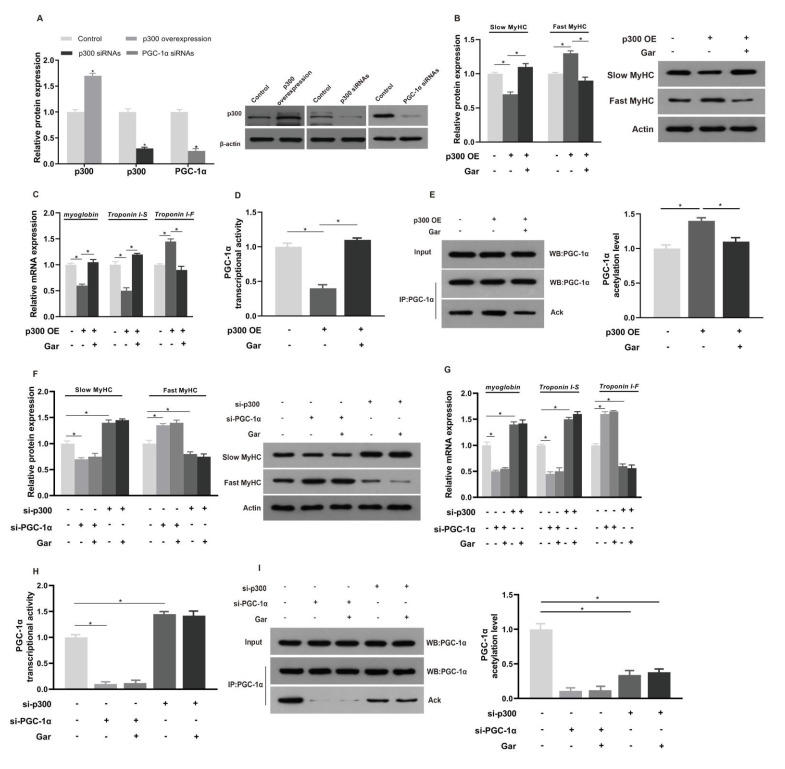
Effects and mechanisms of dietary garcinol on the protein expression of p300 (**A**,**B**), Slow MyHC, Fast MyHC (**C**,**D**), the activity of PGC-1α (**E**), the acetylation level of PGC-1α (**F**) in C2C12 myotubes. The mRNA expression of myoglobin, Troponin I-S, Troponin I-F (**G**), the activity of PGC-1α (**H**), the acetylation level of PGC-1α (**I**) in C2C12 myotubes were transfected with si-p300 or si-PGC-1α. Data are shown as the mean ± SE, n = 10. * Different from the control group, *p* < 0.05.

**Table 1 ijms-24-02702-t001:** Effect of dietary garcinol supplementation on antioxidant status of serum, TA and soleus muscle in mice ^1^.

Item	Control	Low Gar	Mid Gar	High Gar	*p*-Value
Serum					
SOD (U/mL)	39.6 ± 0.96 ^b^	42.46 ± 1.03 ^b^	49.37 ± 0.95 ^a^	52.43 ± 1.35 ^a^	<0.05
CAT (U/mL)	1.21 ± 0.23 ^b^	1.35 ± 0.14 ^b^	2.25 ± 0.17 ^a^	2.46 ± 0.13 ^a^	<0.05
GSH (μmol/L)	3.28 ± 0.35 ^b^	3.56 ± 0.27 ^b^	5.26 ± 0.23 ^a^	5.57 ± 0.31 ^a^	<0.05
GPx (U/mL)	220.7 ± 3.16 ^b^	253.6 ± 2.58 ^a^	259.5 ± 5.69 ^a^	263.4 ± 4.75 ^a^	<0.05
MDA (μmol/L)	6.34 ± 0.21	6.24 ± 0.18	6.16 ± 0.24	6.15 ± 0.19	0.35
TA					
SOD (U/mg prot)	145.63 ± 5.26 ^b^	152.68 ± 4.68 ^b^	185.34 ± 3.26 ^a^	188.74 ± 4.71 ^a^	<0.05
CAT (U/mg prot)	1.26 ± 0.09 ^b^	1.38 ± 0.12 ^b^	2.36 ± 0.07 ^a^	2.54 ± 0.09 ^a^	<0.05
GSH (μmol/g prot)	35.69 ± 2.36 ^b^	38.48 ± 1.28 ^b^	46.85 ± 3.26 ^a^	48.42 ± 2.27 ^a^	<0.05
GPx (U/mg prot)	18.63 ± 1.96 ^b^	20.45 ± 1.15 ^b^	26.85 ± 1.58 ^a^	28.63 ± 2.13 ^a^	<0.05
MDA (nmol/mg prot)	6.32 ± 0.17 ^a^	6.15 ± 0.09 ^a^	5.54 ± 0.11 ^b^	5.14 ± 0.08 ^b^	<0.05
Soleus					
SOD (U/mg prot)	176.95 ± 6.34 ^b^	185.64 ± 4.25 ^b^	223.64 ± 5.12 ^a^	245.69 ± 4.37 ^a^	<0.05
CAT (U/mg prot)	1.45 ± 0.14 ^b^	1.53 ± 0.08 ^b^	2.24 ± 0.12 ^a^	2.45 ± 0.13 ^a^	<0.05
GSH (μmol/g prot)	42.63 ± 2.24 ^c^	50.24 ± 1.25 ^b^	68.53 ± 2.36 ^a^	70.46 ± 3.24 ^a^	<0.05
GPx (U/mg prot)	21.36 ± 1.96 ^b^	24.56 ± 2.17 ^b^	33.67 ± 2.03 ^a^	36.42 ± 2.17 ^a^	<0.05
MDA (nmol/mg prot)	5.53 ± 0.08 ^a^	5.42 ± 0.07 ^a^	4.56 ± 0.11 ^b^	4.16 ± 0.12 ^c^	<0.05

^1^ Values are means ± SE, n = 10/group. Labeled means in a row without a common superscript letter differ, *p* < 0.05.

## Data Availability

The datasets used and/or analysed during the current study are available from the corresponding author upon reasonable request.

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
