# Peer review of "Garcinol Promotes the Formation of Slow-Twitch Muscle Fibers by Inhibiting p300-Dependent Acetylation of PGC-1α"

_ijms, 2023, doi:10.3390/ijms24032702_

Round 1

Reviewer 1 Report

Yao et al. investigated the effects of Garcinol supplementation on skeletal muscle performance and fiber-type composition of mice. They report that the polyphenol Garcinol given for 12 weeks improved skeletal muscle strength and exercise endurance (swimming and treadmill exercises) of C57/Bl6J mice. The enhanced exercise endurance capacity can be explained by the increased expression of slow-twitch muscle proteins such as myosin heavy chain I, troponin I, and Myoglobin. Mechanistically, Garcinol suppresses the activity of p300, an inhibitor of PGC1a expression. Therefore, Soleus and TA skeletal muscles exhibited higher PGC1a expression and activity, and greater activities of Succinate and Malate dehydrogenases. In vitro experiments using differentiated C2C12 myotubes confirmed that the p300-PGC1a pathway is required for Garcinol-induced fast-to-slow muscle phenotype conversion. Overall, the studies are well-designed and the results advance the understanding of Garcinol supplementation and skeletal muscle function and phenotype.

Major comments

It is now accepted that skeletal muscle fibers are hybrids and the predominance of the expression of certain proteins, mainly myosin heavy chain isoforms, denotes their type (slow oxidative vs fast oxidative vs fast glycolytic). Since the authors performed whole tissue mRNA/protein extraction it is not proper to state that Garcinol treatment decreased fast fibers and increased slow fibers (page 3). The current data makes it impossible to say that the fiber type per se changed. For instance, the authors should only strengthen that Garcinol upregulated the expression of Myh1 and downregulated Myh2. Please, reconcile accordingly including Fig 7.

            If the authors really want to confirm whether the fiber type actually changed, the authors will need to phenotype the muscle fibers individually (e.g. by histology).

The I, IIa, IIx and IIb nomenclature denotes protein while MyHC7, 1, 2 and 4 denote the genes. Please, modify Figure 2 accordingly.

The last paragraph of the Introduction section is misleading/confusing: a) Garcinol effects are poorly explained; b) Why does Garcinol is a natural inhibitor? ; c) The sentence in line 64 is confusing; d) the purpose of the study came after results; e) you might want to add a conclusion sentence   

How Garcinol doses were chosen?

Minor comments

Please, define ADG, ADFI, F/G (line 80) and KM (line 85) as they first appear in the results section. Is it correct to say “Average daily feed intake” instead of “food intake”?

When describing the results obtained with C2C12, please strengthen that they were differentiated into myotubes.

Please, avoid long sentences, especially to describe the results. Rephrase the sentence starting in line 159.

Please, avoid dubious words such as “could be upregulated” (line 181), and “may promote” (line 248). In addition, it does not sound right to begin the conclusion with “In general, …”

The conclusion must be based on the results obtained.

Author Response

Reviewer 1

Yao et al. investigated the effects of Garcinol supplementation on skeletal muscle performance and fiber-type composition of mice. They report that the polyphenol Garcinol given for 12 weeks improved skeletal muscle strength and exercise endurance (swimming and treadmill exercises) of C57/Bl6J mice. The enhanced exercise endurance capacity can be explained by the increased expression of slow-twitch muscle proteins such as myosin heavy chain I, troponin I, and Myoglobin. Mechanistically, Garcinol suppresses the activity of p300, an inhibitor of PGC1a expression. Therefore, Soleus and TA skeletal muscles exhibited higher PGC1a expression and activity, and greater activities of Succinate and Malate dehydrogenases. In vitro experiments using differentiated C2C12 myotubes confirmed that the p300-PGC1a pathway is required for Garcinol-induced fast-to-slow muscle phenotype conversion. Overall, the studies are well-designed and the results advance the understanding of Garcinol supplementation and skeletal muscle function and phenotype.

 Response: Thank you very much for your recognition of our study. We have made corresponding changes in the revised version according to your suggestions.

Major comments

It is now accepted that skeletal muscle fibers are hybrids and the predominance of the expression of certain proteins, mainly myosin heavy chain isoforms, denotes their type (slow oxidative vs fast oxidative vs fast glycolytic). Since the authors performed whole tissue mRNA/protein extraction it is not proper to state that Garcinol treatment decreased fast fibers and increased slow fibers (page 3). The current data makes it impossible to say that the fiber type per se changed. For instance, the authors should only strengthen that Garcinol upregulated the expression of Myh1 and downregulated Myh2. Please, reconcile accordingly including Fig 7. If the authors really want to confirm whether the fiber type actually changed, the authors will need to phenotype the muscle fibers individually (e.g. by histology). The I, IIa, IIx and IIb nomenclature denotes protein while MyHC7, 1, 2 and 4 denote the genes. Please, modify Figure 2 accordingly.

Response: Thank you for making such valuable suggestions after reading the manuscript carefully. We have made corresponding changes in the revised version according to your suggestions. We feel that Fig7 does overstate the conclusion and we have removed it.

The last paragraph of the Introduction section is misleading/confusing: a) Garcinol effects are poorly explained; b) Why does Garcinol is a natural inhibitor? ; c) The sentence in line 64 is confusing; d) the purpose of the study came after results; e) you might want to add a conclusion sentence   

Response: Thank you very much for giving the manuscript such a carefully review. We have made corresponding changes in the revised version according to your suggestions.

How Garcinol doses were chosen?

Response: Thank you very much for giving the manuscript such a carefully review. According to the results of previous experiment, the dose-response curve of garcinol was determined in the supplementary materials (Figure S1). Treatment of 0-30 μM garcinol in C2C12 myotubes had no significant effect on cell viability, so we choose 20 μM as the treatment concentration.

Minor comments

Please, define ADG, ADFI, F/G (line 80) and KM (line 85) as they first appear in the results section. Is it correct to say “Average daily feed intake” instead of “food intake”?

Response: Thank you for making such valuable suggestions after reading the manuscript carefully. We have made corresponding changes based on your suggestions.

When describing the results obtained with C2C12, please strengthen that they were differentiated into myotubes.

Response: Thank you very much for giving the manuscript such a carefully review. We have made corresponding changes based on your suggestions.

 Please, avoid long sentences, especially to describe the results. Rephrase the sentence starting in line 159.

Response: Thank you for making such valuable suggestions after reading the manuscript carefully. We have made corresponding changes based on your suggestions.

Please, avoid dubious words such as “could be upregulated” (line 181), and “may promote” (line 248). In addition, it does not sound right to begin the conclusion with “In general, …” The conclusion must be based on the results obtained.

Response: Thank you very much for giving the manuscript such a carefully review. We have made corresponding changes based on your suggestions.

Reviewer 2 Report

This study examined the effects of garcinol on the transition of slow to fast fibers and its associations with p300 and PGC-1α. The authors clearly demonstrated that garcinol increased slow fibers and decreased fast fibers due to reduced p300 activity and increased PGC-1a acetylation. The data are clear and the conclusion is supported by the results. For the publication,  I require several revisions throughout the manuscript.

All miner comments

Overall

1.      Why authors used TA and soleus muscles? Why authors started from 3-wk aged mice? Authors should explain elsewhere.

2.      Authors should define the word when the word appeared first time (PGC-1aADGADFIKM mice?)

Introduction

3.      In the Introduction, I felt that it is better to add more explanation about garcinol, which is not a famous nutrient.

Results and Discussion

4.      Describe the reason why antioxidant enzyme activities were measured in the blood (Table1).

5.      Authors demonstrated grip strength increased but slow fibers also increased. I estimated more slow fibers would decrease grip strength. Could you explain the interpretation of these data in the Discussion.

6.      Line 100, 101, the sentence “while the Mid and High groups had higher (P<0.05) expression levels than the control and Low Gar groups”, must be LOWER, not higher.

7.      In Fig.6, could you add quantitative data, not only band images, about acetylation (Fig6E and I), if possible. Also, add why PGC-1a protein content was not altered even in si-PGC1a cells.

8.      Overall, the figure image resolution is low. In particular Fig.4F and G, I cannot see significant symbols.

9.      Line192-195, the Authors described that “The increase of fatigue-resistant may be related to the decrease of oxidative stress”. The effect of antioxidant supplements on fatigue-resistant is still controversial (Mason et al. 2022 Redox Biol. PMID: 32127289). I recommend discussing mostly the importance of increases in slow fibers induce endurance capacity in this study. There is also evidence that antioxidant enzymes are more in slow fibers than fast fibers (Picard et al. 2012 Am J Physiol Cell Physiol. PMID: 22031602). This evidence would be a reason why muscle antioxidant enzyme activities increased after garcinol supplementation.

Methods

10.    In the method, when did the authors do grip, swimming and treadmill tests? The authors should describe when these tests were done.

11.    In 5.6. Enzymatic activity assay, only C2C12 was explained. TA and soleus muscles were also measured for these enzyme activities. The authors should describe how to measure these enzyme activities in these muscles. In particular, I would like to know how to homogenize the muscle before using the kit.

12.    In 5.9. Antioxidant enzyme activity analysis, catalog numbers should be added.

References

13.    The reference list should be written with the same description.

Author Response

Reviewer 2

This study examined the effects of garcinol on the transition of slow to fast fibers and its associations with p300 and PGC-1α. The authors clearly demonstrated that garcinol increased slow fibers and decreased fast fibers due to reduced p300 activity and increased PGC-1a acetylation. The data are clear and the conclusion is supported by the results. For the publication,  I require several revisions throughout the manuscript.

Response: Thank you very much for your recognition of our study. We have made corresponding changes in the revised version according to your suggestions.

All miner comments

Overall

  1. Why authors used TA and soleus muscles? Why authors started from 3-wk aged mice? Authors should explain elsewhere.

Response: Thank you very much for giving the manuscript such a carefully review. The tibialis anterior (TA) and soleus muscles are representative of the fast and slow twitch muscle types, respectively. Skeletal muscle development was rapid in the 3-week-aged mice, and our experiment lasted for 12 weeks.

  1. Authors should define the word when the word appeared first time (PGC-1a,ADG,ADFI,KM mice?)

 Response: Thank you very much for giving the manuscript such a carefully review. We have made corresponding changes based on your suggestions.

Introduction

  1. In the Introduction, I felt that it is better to add more explanation about garcinol, which is not a famous nutrient.

Response: Thank you for making such valuable suggestions after reading the manuscript carefully. We have made corresponding changes in the revised version according to your suggestions.

Results and Discussion

  1. Describe the reason why antioxidant enzyme activities were measured in the blood (Table1).

Response: Thank you very much for giving the manuscript such a carefully review. Garcinol is an excellent antioxidant nutrient. Meanwhile, studies have shown that the more slow muscle fibers, the greater the antioxidant capacity. Therefore, we tested the activity of antioxidant enzymes in blood.

  1. Authors demonstrated grip strength increased but slow fibers also increased. I estimated more slow fibers would decrease grip strength. Could you explain the interpretation of these data in the Discussion.

Response: Thank you for making such valuable suggestions after reading the manuscript carefully. To further investigate the effects of garcinol on skeletal muscle contraction properties, we first tested the exercise capacity of mice. We found that garcinol dose-dependently increased muscle grip strength (Fig 1F), swimming time (Fig 1G ), low-speed running time (Fig 1H). However, high-speed running time was unchanged by garcinol supplementation (Fig 1I), which indicates garcinol may specifically improve endurance exercise performance, but not explosive exercise performance.

  1. Line 100, 101, the sentence “while the Mid and High groups had higher (P<0.05) expression levels than the control and Low Gar groups”, must be LOWER, not higher.

Response: Thank you very much for giving the manuscript such a carefully review. We are sorry for the mistake. We have made the corresponding changes in revised version.

“while the Mid and High groups had lower (P<0.05) expression levels than the control and Low Gar groups”

  1. In Fig.6, could you add quantitative data, not only band images, about acetylation (Fig6E and I), if possible. Also, add why PGC-1a protein content was not altered even in si-PGC1a cells.

Response: Thank you very much for giving the manuscript such a carefully review. We have supplemented the data on the quantification of acetylation in Fig 6. PGC-1a protein content was decreased in si-PGC1a cells (Fig 6A).

  1. Overall, the figure image resolution is low. In particular Fig.4F and G, I cannot see significant symbols.

Response: Thank you very much for giving the manuscript such a carefully review. We have updated the images in high resolution.

  1. Line192-195, the Authors described that “The increase of fatigue-resistant may be related to the decrease of oxidative stress”. The effect of antioxidant supplements on fatigue-resistant is still controversial (Mason et al. 2022 Redox Biol. PMID: 32127289). I recommend discussing mostly the importance of increases in slow fibers induce endurance capacity in this study. There is also evidence that antioxidant enzymes are more in slow fibers than fast fibers (Picard et al. 2012 Am J Physiol Cell Physiol. PMID: 22031602). This evidence would be a reason why muscle antioxidant enzyme activities increased after garcinol supplementation.

Response: Thank you for making such valuable suggestions after reading the manuscript carefully. We have made corresponding changes based on your suggestions.

Methods

  1. In the method, when did the authors do grip, swimming and treadmill tests? The authors should describe when these tests were done.

Response: Thank you very much for giving the manuscript such a carefully review. Thirty minutes after the last treatment, mice were made to perform a endurance and strength exercises.

  1. In 5.6. Enzymatic activity assay, only C2C12 was explained. TA and soleus muscles were also measured for these enzyme activities. The authors should describe how to measure these enzyme activities in these muscles. In particular, I would like to know how to homogenize the muscle before using the kit.

Response: Thank you very much for giving the manuscript such a carefully review. We have made corresponding changes based on your suggestions.

The lactate dehydrogenase (LDH, Catalog No. A020-2-1) activity, malate dehydrogenase (MDH, Catalog No. A021-2-1) activity, and succinic dehydrogenase (SDH, Catalog No. A022-1-1) activity in the TA muscle and soleus muscle of mice and in C2C12 myotubes were measured using commercial kits according to the manufacturers’ instructions (Nanjing Jiancheng Bioengineering Institute, Nanjing, China). Complex III ELISA kit and Complex IV ELISA kit (Shanghai Enzyme-linked Biotechnology Co., Ltd, Shanghai) were used to test Complex III and Complex IV concentrations in muscle tissue.

  1. In 5.9. Antioxidant enzyme activity analysis, catalog numbers should be added.

 Response: Thank you very much for giving the manuscript such a carefully review. We have made corresponding changes based on your suggestions.

References

  1. The reference list should be written with the same description.

Response: Thank you very much for giving the manuscript such a carefully review. We have made corresponding changes based on your suggestions.
